# Optimized Technologies for Cointegration of MOS Transistor and Glucose Oxidase Enzyme on a Si-Wafer

**DOI:** 10.3390/bios11120497

**Published:** 2021-12-05

**Authors:** Cristian Ravariu, Catalin Corneliu Parvulescu, Elena Manea, Vasilica Tucureanu

**Affiliations:** 1BioNEC Group, Department of Electronic Devices Circuits and Architectures, Polytechnic University of Bucharest, Splaiul Independentei 313, 060042 Bucharest, Romania; 2National Institute for Research and Development in Microtechnologies, 077190 Voluntari, Ilfov, Romania; elena.manea@imt.ro (E.M.); vasilica.tucureanu@imt.ro (V.T.)

**Keywords:** ENFET technology, nanostructured TiO_2_, bionanomicrotechnologies, engineering of surfaces

## Abstract

The biosensors that work with field effect transistors as transducers and enzymes as bio-receptors are called ENFET devices. In the actual paper, a traditional MOS-FET transistor is cointegrated with a glucose oxidase enzyme, offering a glucose biosensor. The manufacturing process of the proposed ENFET is optimized in the second iteration. Above the MOS gate oxide, the enzymatic bioreceptor as the glucose oxidase is entrapped onto the nano-structured TiO_2_ compound. This paper proposes multiple details for cointegration between MOS devices with enzymatic biosensors. The Ti conversion into a nanostructured layer occurs by anodization. Two cross-linkers are experimentally studied for a better enzyme immobilization. The final part of the paper combines experimental data with analytical models and extracts the calibration curve of this ENFET transistor, prescribing at the same time a design methodology.

## 1. Introduction

Enzyme field effect transistors (ENFETs) are electronic devices that integrate in the same chip a FET (field effect transistor) with a sensitive enzymatic membrane for biorecognition purposes [1]. ENFET fabrication is a complex interdisciplinary technology related to microelectronics [2], biochemistry [3], nanomaterials for biostructures, and biocompatibility [4]. On the other hand, metal oxide semiconductor (MOS) downscaled technology has just reached its ultimate stage. As predicted by Moore [5], in 2021, we have just assisted in the glorious 2-nm node launching [6]. Predictions about the post-MOS era were issued 10 years ago. In 2011, the IEEE Chairman of the Electron Devices Society from Europe proposed the MOS diversification by its cointegration with other nanobiomaterials [7]. Many researchers have devoted large studies to biosensor cointegration, mainly with carbon nanotube CNT-FET or graphene FETs, especially for the femtomolar detection limit [8,9]. These solutions may be successful in the industry in 15 years, when the price/quality ratio of carbon-related technologies will equal the excellent reproducibility and price of the MOS-FET transistors. Therefore, the nanoelectronics roadmap still emphasizes that biosensors with MOS transducers will be the first beneficiaries of the cointegration between the MOS part and enzymatic receptor [10].

This adherence is one of the critical issues for the enzyme immobilization onto Si-wafers [11]. The proposed solutions combine crosslinked chemical bonds with physical adsorption methods [12]. Another optimized technique consists of enzyme immobilization on Si-porous, directly converted on the surface of the Si-wafer [13]. For instance, Si-porous was used to entrap an acetylcholine-esterase enzyme for a pesticides biosensor, with the advantage of Si-compatible technology [14]. Another issue in this discussion is the optimal electrodes selection. However, Al electrodes for source/drain contacts are separately configured, requiring additional costs. Titanium is an alternative candidate for electrodes that touch the biological probes. It was recently used in a glucose biosensor [15]. Another research group reported a transistorless glucose biosensor with a TiO_2_-binding layer. It was based on the immobilization of the glucose oxidase (GOx) enzyme by chitosan onto titanium dioxide nanotube arrays. The GOx–chitosan/TiO_2_ biosensor showed a sensitivity of 5.46 μA·mM^−1^ with a linear range from 0.3 to 1.5 mM [16]. Another amperometric glucose biosensor, still transistorless, used a GOx receptor bound to hydrogen titanate nanotubes by crosslinking to form a working electrode. The fabricated GOx–Ti electrode had a linear response to 1–10 mM of glucose, a sensitivity of 1.541 μA·mM^−1^ per cm^2^, and a detection limit of 59 μM [17]. A technological solution to produce TiO_2_ nanowires on a solid-state surface can be thermal oxidation for chemical gas sensing [18] or anodization for a glucose sensor [15]. By cointegration of the GOx enzyme with a flexible organic transistor, another glucose detector was produced [19]. A drain current increase was observed from 1 μA (in the absence of glucose) to 4 μA (in the presence of glucose) at V_G_ = 3 V. However, the organic structure was a FET but not a MOS-FET. The silicon cointegration purpose was not yet achieved. Therefore, GOx immobilization on nanostructured materials onto Si-wafers, together with the configuration of the Ti electrodes, under the stress of the area delimitation by lithography still rusts, a serious challenge for MOS-FET transistors.

The ENFET from this paper is a glucose biosensor that contains a modified MOS-FET transistor fabricated on Si-wafer and a glucose oxidase-sensitive membrane. By a separate experimental study, we investigated the nanostructured TiO_2_ synthesis and the sensitive enzymatic membrane deposition on a Si-wafer. The following targets were exceeded versus the previous states [15,16,17,18,19]: (i) the fabrication steps of the ENFET include a channel conductivity correction, (ii) nanostructured Al_2_O_3_ [20] or Si-porous [21] films are replaced by nanostructured TiO_2_ films, (iii) nanostructured TiO_2_ films and GOx membranes are characterized by SEM and FTIR, (iv) GOx membrane immobilization is optimized by alternative crosslink agents, and (v) a calibration curve is provided, and it is accompanied by an experimental model.

## 2. Method

### 2.1. FET Part Fabrication

In this section, the technological flow of the FET part from ENFET is presented. A Si-type wafer <100> doped by Boron 10^15^ cm^−3^, with a 3-inch width and 450-μm thickness, starts the process. The chemical reagents usually used in microelectronics were purchased from CheMondis, while the enzymes and crosslinkers were acquired from a Sigma provider with a purity grade between 96 and 99.99%. The main processes occurred in a clean room and part of processes in grey rooms (certified ISO 6&5-200 sqm, ISO7-120 sqm, and ISO 8-300 sqm) [22]. The micro- and nanolithography processes appealed at the mask shop, besides the laser lithography system-DWL 66 fs (Heidelberg Instruments Mikrotechnik, Heidelberg, Germany), double-sided mask aligner MA6/BA6-Suss MicroTec, Germany. The layer depositions were produced by Electron Beam Evaporation TEMESCAL FC-2000 (Temescal, Livermore, CA, USA) or nanoengineering workstation-e-Line-Raith GmbH, Germany, resolution line 10 nm. The entire MOS-FET process occurred in IMT-MINAFAB infrastructure that was SR EN ISO 9001:2008 certified by TÜV Thüringen, Germany [22].

The FET technological steps are available in Figure 1. In the preliminary stage, the wafers were cleaned in H_2_SO_4_ (96%):H_2_O_2_ (30%) (3:1) solution, followed by HF (10%) immersion to remove the native contaminated oxide. After deionized water washing, the wafers were dried in N_2_ atmosphere. The next step was devoted to a thermal mask oxide grown on the front side and inherently on the bottom side of the wafer (Figure 1a). In the next step, the oxide was completely etched from the FET area. In the opened window, a thin 30-nm oxide was grown as the preimplantation film (Figure 1b). Through this thin oxide, the Boron implantation produced external p^+^-wells, and Arsenic (As) implantation produced n^+^-wells for the source/drain zones (Figure 1c–e). The ionic implantation process was characterized by doses of 5 × 10^15^ ÷ 10^16^ at/cm^2^ and energies of 50 ÷ 125 keV.

After the implantation, the subsequent annealing stages at a maximum 1100 °C temperature in oxidant atmosphere allowed impurity redistribution. In this way, the p^+^-well rings reached 6.9-μm depth, and the n^+^-drain/source regions reached 2.4-μm depth (Figure 1e). The As diffusion was done in oxidizing atmosphere in order to simultaneously achieve a diffusion of the impurities and growing of an oxide layer, which was subsequently used as a mask for the gate oxide. An oxide of 380 nm was increased during the diffusion process. The thickness of the oxide was measured by the ellipsometric method. The diffusion depths and V/I were measured on a control wafer that followed the batch.

Before the gate oxide configuration, a channel conductivity correction was performed. A new window was opened in the oxide to the Si-surface. Then, a preimplantation thin oxide was grown as sacrificial oxide. A correction channel implant was produced by a dose of As, 2 × 10^12^ at/cm^2^ (Figure 1f). Due to the sacrificial oxide contamination, this one was eliminated in a buffer solution of BHF 6:1. The gate oxide configuration was preceded by a succession of cleanings in different acids to de-contaminate the silicon surfaces of any organic or ionic compounds within the gate window.

Then, a gate oxide of was created 25 nm by dry oxidation in an extremely clean environment, with the O_2_ atmosphere at 1000 °C. Optionally, over oxide, a nitride layer was deposited at low pressure by the LPCVD process (T = 750 °C; *p* = 330 mTorr, t = 20 min) on the entire structure (Figure 1g). For metallic contacts, Ti/Au (20 nm/800 nm) films were deposited onto the entire surface by high-vacuum evaporation. The source, drain, and p-well metallic contacts were subsequently configured by the metal mask utilization and metallic rust removal by wet etching (Figure 1h). A metal alloying secondary step was used: annealing at 250 °C for 20 min to decrease the contact resistance between Ti/Au electrodes and the Si-surface and to improve the metal adherence to the semiconductor.

### 2.2. Ti Conversion in Nanostructured TiO_2_ on Si-Wafer

The second validation envisaged the nanostructured TiO_2_ conversion in the gate space. A thin Ti metallic film of 90 nm was deposited above the gate oxide/nitride. This deposition was achieved by sputtering, based on an evaporation process in high-vacuum conditions. Then, the Ti film was converted into a nanostructured TiO_2_ by anodization using an electrochemical combine (Figure 2a). The electrochemical cell contained three electrodes: a Pt cathode (7 × 10 cm^2^ area), a reference calomel electrode (Hg/HgCl_2_/sol. KCl), and a mechanical device for wafers clamping as the anode. The anode and cathode electrodes stayed at a 2 ÷ 4 cm distance in an aqueous solution of oxalic acid and phosphoric acid as an electrolyte. A Hameg source provided voltage variations between electrodes from 0 V to 5 V and then to 30 V at a rate of 0.05 ÷ 0.6 V/s, keeping the optimal pH at 6.8, by a phosphate buffer.

The decreasing of the electrochemical current toward zero indicated the anodization finishing (Figure 2b). After anodization, the TiO_2_ structures were amorphous. The subsequent steps were followed to convert it into a crystalline form. The final structures were intensively cleaned in solvent, then washed in deionized water and dried in alcohol vapors. Next, an annealing step was applied. The optimal post-anodization annealing occurred at 450 °C for 45 min in an N_2_ environment. This allowed the final nanostructured TiO_2_ film to take an anatase crystalline form and to be strongly anchored to the Si-wafer. The characterization of this film is included in Section 3.

### 2.3. GOx Enzyme Immobilization

The next step was devoted to the fabrication of the biosensitive enzymatic membrane above the gate space. To initiate this step, the sensitive area was configured by a photopoligraphic technique, opening a window into the gate area to entrap GOx. In a previous work, we used nafion as the crosslinking agent [11]. The enzymatic film homogeneity was scanty, presenting crashes and poor uniformity. In this paper, new variants are experimentally tested for 2 crosslinkers: (i) glutaraldehyde and (ii) glutaraldehyde + nafion—both in a complete process. The benefits of the last combination were good mechanothermal properties from nafion and proper adherence from glutaraldehyde. Glucose oxidase (GOx), glutaraldehyde (GA), and nafion (NA) were from Sigma.

In order to optimize the enzyme adhesion to the nanostructured TiO_2_ film, a sialylation treatment for 2 min in 3-aminopropyltriethoxysilane, followed by annealing at 900 °C for 30 min, occurred for both crosslinker variants. The GOx enzyme was solubilized in 0.002-mM phosphate buffer, pH = 7.4. GOx 6.6 mg was dissolved in 66-μL PBS (phosphate-buffered saline). Then, in case (i), 20-µL GA 1% was mixed with 20 µL of previous GOx solution. In case (ii), 5-µL GA and 5-µL nafion were mixed with 5 µL of the previous GOx solution. Then, the membrane stayed at room temperature for polymerization for 120 min; after which, it was washed in 0.002-mM phosphate buffer. Between measurements, the ENFETs were stored in a refrigerator at 4 °C. Before the beginning of the experiment, the sensors were kept 2 h at room temperature for balancing. The results will be presented in Section 3 and Section 4.

### 2.4. Final ENFET

During the last technological step, the ENFET structures were separated, and the usual encapsulation techniques were applied in order to offer the final ENFET device. The encapsulation support was textolit plated by copper. In each chip, an identical MOS-FET was cointegrated with an ENFET. The ENFET structure is presented in Figure 3a. The GOx layer was moistened by an aqueous drop containing glucose, and it was contacted by the gate through a reference electrode. The usual reference electrode was carried out by Ag/AgCl solid-state material. A challenging issue was to produce solid-state reference electrodes in integrated ENFETs, compatible with MOS-FET technology [23]. In order to ensure the silver adhesion, a Ti layer of 50 nm was firstly deposited onto the Si surface in a specially delimited area, according to Figure 3a. There, distinct color codes suggest different materials. Then, the silver layer was deposited at 400-nm thickness with a rate of 40 Ǻ/s. The final reference electrode with AgCl solid-state material was performed by selective chlorination through a photo-resistant mask of an Ag film of 100-nm thickness. The Ag film deposition in high vacuum at a controlled rate must ensure the adherence of the AgCl to the initial Ag substrate. The entire process occurred into a vacuum deposition installation with electron beams at a pressure under 6 · 10^−6^ Torr. The theory of the AgCl species forming during the chlorination process was previously depicted [24].

Figure 3b presents the measured transfer characteristics of the fabricated transistor in the saturation regime at V_DS_ = 1 V and in the linear regime at V_DS_ = 0.1 V.

The linear regime induced more linearity of the calibration curve but suffered from low currents. Hence, the saturation regime was preferred.

### 2.5. Microphysical Characterization

The nanostructured films and samples were characterized by a spectrometer FTIR Bruker Tensor 27 and QUANTA INSPECT F scanning microscope (SEM) imaging tool.

In accordance with the XR spectra, the ATR module with KRS 5 crystal of the FTIR Tensor 27 spectrometer was used at 45° and 25 reflections in the spectral domain of 4000–370 cm^−1^. After 64 scans, with a resolution of 4 cm^−1^ for the analyzed TiO_2_ layers, the results were well-confirmed.

## 3. Technological Characterizations

In this section, some characterization results from different moments of the ENFET manufacturing process are picked and discussed.

### 3.1. Results for the FET Part

For the FET part, some technological intermediary results were analyzed by microscopy methods. After the impurity diffusions in the p^+^-wells and n^+^-drain/source regions, in a broken Si sample, the depths of these diffusions are estimated in Figure 4a,b.

A strip of the Si sample was glued to the diffused part upwards; then, it was micropolished using an abrasive paste (alumina) on a perfect flat table at an angle of 2° to obtain the section of the diffused area as large as possible. The eroded sample was stained in the diffused area by HF 50% light solution in a dark grey color. The reading of this area was done under a metallographic optical microscope with the eyepiece setting at y_p_∙tg12°, where y_p_ was the hypotenuse length. In this way, a p-well depth of 6.9 μm was measured from this sample (Figure 4a). Similarly, a n^+^-drain/source depth of 2.4 μm was measured (Figure 4b).

As a second validation, the source and drain metallization were emphasized. In Figure 5a, the local source electrode deposition plus alignment signs are visible, while, in Figure 5b, the gold pad for the source is visible. Remember that the top metal was gold for the source and drain, because the Au pads must be compatible with the wire soldering machine from the microelectronics factory.

The third validation concerned the final FET part configuration before Ti film deposition, when the metals contacted the active area of the device.

At this stage, the metal was removed by wet gravure, so that the Ti/Au traces ensured only the source, drain, and gate (Figure 6a). A distinct image of the fabricated FET structures with different colors for different layers is available in Figure 6b.

Then, the Ti film was deposited above through a correspondent mask, followed by TiO_2_ conversion in nanostructured-TiO_2_.

### 3.2. Results for Nanostructured TiO_2_

The samples were characterized by spectrometer FTIR and SEM tools. The best results for the anodizing process were achieved for a potential of 5–10 V, keeping the pH in a range of 6.5–7.3 and preferably slightly acidic. The amount of water in the electrolyte must be high enough to obtain a good dispersion of NH_4_F salt in the cations and anions. The maximum limit of the partial concentration occurred for a saturated solution of NH_4_F/H_2_O 40%. The nanostructured TiO_2_ film at a temperature of 450 °C was the anatase structure, while, at 800 °C, it became rutile.

The SEM image of the TiO_2_ film obtained by optimal anodizing and treated at a temperature of 450 °C in nitrogen atmosphere was visible in a lateral section for the anatase form of nanostructured TiO_2_ (Figure 7a).

The XR spectrum of the TiO_2_ film obtained by optimal anodizing and similar treatment confirmed the anatase form of the nanostructured layer, as can be seen in Figure 7b.

The spectrometer was used to study the chemical bond configuration in film samples by Fourier-transform infrared (FTIR) spectrometry. The spectra were plotted in the wavenumbers 4000–400 cm^−1^ by averaging 64 scans and with a resolution of 4 cm^−1^ at room temperature using an ATR holder with KRS-5 crystal, 25 reflections, and a radiation entry angle at 45°. For better visibility, the CO_2_ spectra bands in domain 2400–2200 cm^−1^ were removed (Figure 8). In a previous publishing, the ATR-FTIR spectrum of untreated TiO_2_ film revealed more diffuse and larger films [17]. Now, cleaner and steeper interfaces are produced. Spectrum 8(a) shows only low-intensity absorption bands that can be attributed to the silicon substrate, bands centered at about 1110 cm^−1^ (Si-O) and 611 cm^−1^ (Si-Si). The Ti film showed no bands in IR. After anodic oxidation of the Ti film and thermal stabilization, in the spectrum of the sample was observed the appearance of characteristic peaks of the nanostructured TiO_2_ film formation, spectrum 8(b). These bands were centered at about 553 cm^−1^ and 494 cm^−1^, respectively, which could be attributed to the vibration mode of the Ti-O bonds. The anchoring of the TiO_2_ film of the substrate was confirmed by the increase in intensity of the bands in the spectral range 1250–1050 cm^−1^ that could be attributed to the formation of Si–O–Ti bonds as a result of the formation of a SiO_2_–TiO_2_ phase.

### 3.3. Results for GOx Imobilization

In a previous work, the GOx enzyme was immobilized in a nafion agent [11]. Preliminary tests of glutaraldehyde binding to the TiO_2_ layer were also briefly tested [15]. In both cases, the final bioreceptor membrane presented poor adherence of the GOx to the substrate. In this section, FTIR and SEM characterization techniques were applied for the enzymatic membrane description in cases (i) with glutaraldehyde and (ii) with glutaraldehyde + nafion. In Figure 9, the FTIR results for the case (i) are presented.

Figure 9 shows the spectra for GOx binding to the nanostructured TiO_2_ substrate via the GA layer-in variant (i). In this spectrum, both characteristic peaks of the substrate and aldehyde could be observed. The low-intensity peak centered at about 554 cm^−1^ was associated with the stretching vibration of the Ti–O bonds. The existence of GA at the TiO_2_ surface was confirmed by the appearance of centered bands at 2949 and 2815 cm^−1^ due to the stretching mode of the C–H bonds and 1745 cm^−1^ assigned to C=O bond stretching vibration. The spectrum from Figure 9b confirms the anchoring of the GOx enzyme to the surface of the oxide film via GA. The presence of GOx is confirmed by the appearance of enzyme-specific amide bands: amide I associated with the stretching vibration of C=O bonds (1644 cm^−1^), the amide II band due to the coupling between the stretching vibration of the CN bonds, and bending of the NH bonds with small contributions from the plane deformation of the CO bonds and the DC- and CN-stretching vibrations (1535 cm^−1^). Additionally, in the spectrum of the sample can be observed the specific amide III double bands associated with the combination between NH bond deformation and CN stretching, with small contributions due to CO deformation and CC stretching vibrations (1301 and 1245 cm^−1^). GOx binding to GA can be explained by the appearance of a high intensity peak at 1070 cm^−1^, which can be attributed to the vibration mode of C–O bonds. The absorption band from 1070 cm^−1^ overlaps over the bands associated with the Si–O–Ti bonds in the substrate, but in the spectrum, a low-intensity band could be observed at 555 cm^−1^ that was associated with Ti–O bonds.

For the second variant of GOx enzyme immobilization on titanium dioxide, another crosslinker was used: GA+ NA, case (ii). Figure 10 presents the FTIR analysis in this case.

The spectrum of the sample from Figure 10a corresponds to the GA + NA mixture deposited on the nanostructured TiO_2_ film. It confirmed both the presence of NA and GA. The GA existence was associated with the appearance of the centered peak at 1745 cm^−1^ (C=O), while the bands in the spectral range 800–400 cm^−1^ were associated with Ti–O–C- and C–O–C-type bonds, respectively, due to nafion anchoring to the substrate. As in the spectrum shown in Figure 10, the GOx anchoring to the GA + NA/TiO_2_/SiO_2_/Si substrate is spectrally confirmed by the appearance of amide band I (C=O) at 1649 cm^−1^ and amide II (N-H) at 1550 cm^−1^. The strips at 1216 cm^−1^ and 1153 cm^−1^ may appear due to the overlap of the C–O bonds, which confirmed the GOx anchorage, with the CF_2_ bonds from nafion.

In conclusion, the best GOx anchoring occurred for case (ii). The results were also confirmed by the SEM characterization, where agglomerations and film inhomogeneities were acceptable in the enzymatic membrane with GA + NA crosslinking agents (Figure 11).

In future experiments, the enzyme immobilization can be further improved by direct binding on the electrode surface by covalent Si–S or Si–O covalent chemistry [25,26,27]. In this way, the intermediate nanostructured layer can be completely eliminated, keeping the receptor membrane as close as possible to the silicon surface.

## 4. Calibration Curve

For the calibration curve of this biosensor, a glucose aqueous solution of different concentrations between 100 mM and 0.1 μM is necessary. GOx is a catalyst that accelerates the oxidation of glucose by oxygen. In the presence of O_2_, β-d-glucose is converted into d-glucono-1.5-lactone, which then hydrolyzes into gluconic acid, while oxygen is reduced to hydrogen peroxide. In our experiments the reaction occurs in normal conditions at *p* = 1 atm, T = 20 °C. The potential variations at the reference electrode modulate the electron concentrations inside the inversion channel and, finally, define the drain current (Figure 12a).

Both tested devices, ENFET and cointegrated MOS-FET, have channel lengths L = 30 μm, width W = 200 μm, threshold voltage V_T_ = 0.59 V, specific oxide capacitance C_ox_ = 1.4·10^−7^ F/cm^2^, and aspect ratio k = W/L·μ_n_·C_ox_ of 1.587 mA/V^2^. The calibration curve can be estimated with these parameters, associated with a recent analytical model for ENFET biosensors [28]. For glucose oxidase, the parameters 1/v_max_ and 1/K_M_ can be extracted from Lineweaver–Burk representation [29]: v_max_ = 1.745 mmol/l·s—the maximum reaction rate and K_M_ = 10.5 mmol/l—the Michaelis constant.

In order to raise the calibration curve of the glucose biosensor, eight identical ENFETs were selected to repeat this experiment and to ensure the reproducibility. Then, different dilutions of the aqueous glucose solution were stored in eight recipients for the concentrations 0.0001 mM, 0.001 mM, 0.01 mM, 0.1 mM, 1 mM, 10 mM, 20 mM, and 100 mM. In turn, each transistor was biased in Hameg double power supply HM8012 at constant voltages: V_DS_ and V_GS_ that ensured the saturation. The drain currents in mA were recorded by Hameg curve tracer HM6042, and the sub-1 mA currents were measured by a Keithley 6487 Picoammeter. First, the current was measured in the absence of any analyte. Then, a drop of glucose solution of the lowest concentration was dropped on the enzymatically active area. After complete wetting of the enzyme matrix, we always waited for 2 min to let the enzymatic reaction act. Then, the current was measured. After noting the data, the procedure was repeated for the next glucose concentration. All the data current concentrations were noted.

The calibration curve of the glucose biosensor is provided in Figure 12b, after calculating the recorded average current at each glucose concentration from those eight tested transistors. Each ENFET was tested in two work regimes: (i) saturation at V_DS_ = 2 V and (ii) linear regime at V_DS_ = 0.2 V, keeping V_GS_ = 4 V more than the threshold voltage to ensure a strong inversion regime and a higher current. The linear regime offers poor linearity in Figure 12. The linear range can be approximated only for glucose concentrations sub-10 μM. The saturation regime seems to be the proper work regime for a good linearity in a larger glucose concentration range of −0.001 mM–100 mM—a larger range than the linear range of other glucose biosensors [16] but with a comparable detection limit to other biosensors [17]. Based on the excellent GOx selectivity, in conditions of nanostructured TiO_2_ anchoring [29], we expect to avoid interferences with other -ose analytes, like lactose and fructose.

Aided by an analytical model [30], a sensitivity of 0.43 μA/mM can be extracted for our biosensor that is close to a sensitivity of the glucose biosensor with titanate nanotubes [17]. However, the paper contributes to the technology optimization for integrated biosensors on Si-wafers. Secondly, the narrow linearity encountered when ENFET was operated in a linear regime suggests very good narrow linear ranges at extremely low glucose concentrations. In fact, when ENFET is working in a linear regime as the transistor, it can detect very narrow and low glucose concentrations (e.g., in interval 0.0001–0.0005 mM, we had a good sensitivity around 1.5 μA/0.0001 mM).

The glucose samples always used water as the solvent, keeping the optimal pH to 7.2 by a phosphate buffer. This optimum was extracted from a brief pH sensitivity test. Three glucose samples of 10 mM were prepared, increasing the phosphate concentration from 5 mM to 60 mM, so that three clear pH were recorded by a pH meter: 6.4, 7.2, and 7.7, as the blood pH can vary within these limits. ENFET was biased in the saturation for each measurement. A drop of the first pH solution was dropped on the enzymatically active area. After 2 min of waiting, the drain current was recorded. The procedure was repeated for next pH values at the same room temperature each time. The drain current registered the values 1.55 mA, 1.58 mA, and 1.57 mA for pH equal to 6.4, 7.2, and 7.7, respectively. The measurements were in agreement with those from the literature [31,32]. This experiment indicated to us an optimum pH of 7.2 for the phosphate buffer that ensured the maximum drain current.

The next step was to extend the ENFET tests to controlled biological samples. Similar tests demonstrated the functioning of the glucose biosensors with transistors in artificial sweat media [33] and then in biological media taken either from sweat [34] or blood or saliva [35]. Obviously, these measurements in living environments raise new challenges, such as the harmonization of microelectronic technology for an ENFET with the use of biocompatible materials for in vivo measurements. Some steps have already been taken to detect ions in human perspiration using wearable ISFET transistors [36]. The wearable ENFET stage will follow in the next years.

## 5. Conclusions

In this paper, a detailed technological flow of an enzyme field effect transistor was presented. Some key technological steps for the FET part were gate oxide configuration and channel conduction correction by an ion implantation. Microphysical and chemical characterizations were performed on the nanostructured TiO_2_ film and on the glucose oxidase membrane with two kinds of crosslinkers. The optimal annealing occurred at 450 °C for 45 min in an N_2_ environment that allowed for the final nanostructured TiO_2_ film to be strongly anchored to the Si-wafer.

The FTIR and SEM results demonstrated that the optimal technology to immobilize the GOx membrane was the formula: glucose oxidase + glutaraldehyde + TiO_2_/SiO_2_/Si. The spectrum was clear and well-defined, which confirmed the optimal anchoring of the glucose oxidase to the substrate. Finally, some experiments allowed to raise the calibration curve of this biosensor. If the ENFET transistor was operated in saturation, we benefited on a large linearity range of 0.001–100 mM for glucose at a poor sensitivity of 0.43 μA/mM. If the ENFET was operated in a linear regime at a low V_DS_, the sensitivity drastically increased at extremely low glucose concentrations (1.5 μA/0.0001 mM), with a paid price of an extremely narrow linearity range.

## Figures and Tables

**Figure 1 biosensors-11-00497-f001:**
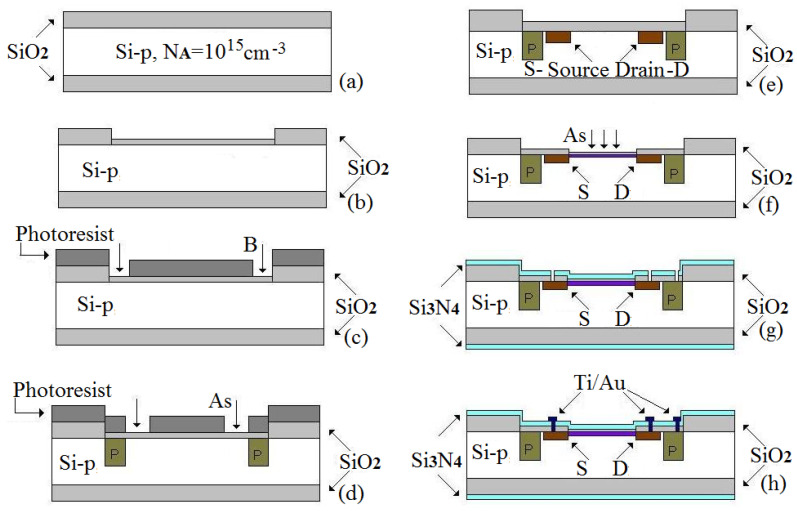
The successive technological steps during FETs processing. (**a**) thermal mask oxide grown (**b**) thin oxide grown in opened window (**c**) Boron implantation for p^+^-wells (**d**) As implantation for n^+^-wells (**e**) annealing stages in oxidant atmosphere (**f**) correction channel implant (**g**) gate oxide realization and nitride layer deposition (**h**) metallic contacts configuration.

**Figure 2 biosensors-11-00497-f002:**
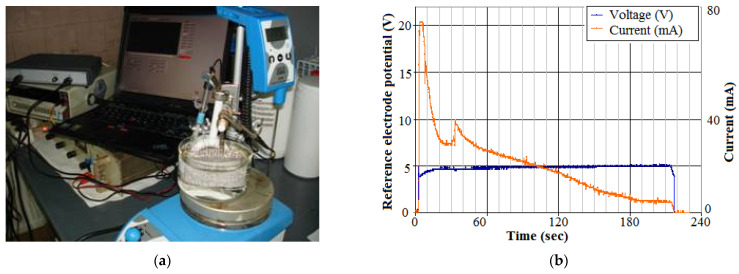
(**a**) The electrochemical combine used for the Ti-film conversion in nanostructured TiO_2_, and (**b**) potential of the reference electrode and current during anodization.

**Figure 3 biosensors-11-00497-f003:**
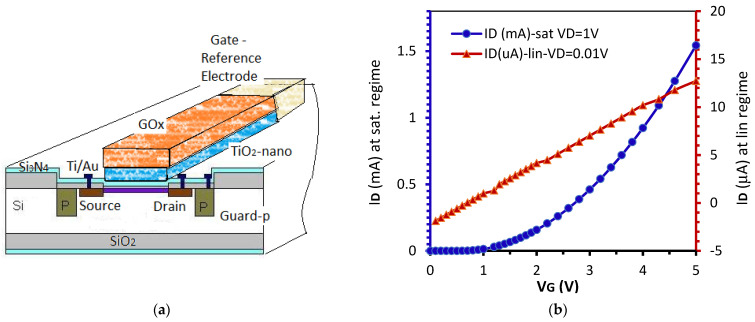
(**a**) Structure of the ENFET glucodetector, and (**b**) measured transfer characteristics in the glucose absence.

**Figure 4 biosensors-11-00497-f004:**
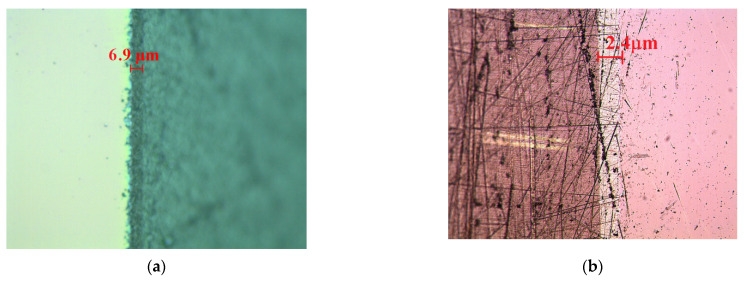
(**a**) Depth of the guard ring (p^+^-well) of 6.9 μm and (**b**) n^+^-well of 2.4-μm depth (light gray area).

**Figure 5 biosensors-11-00497-f005:**
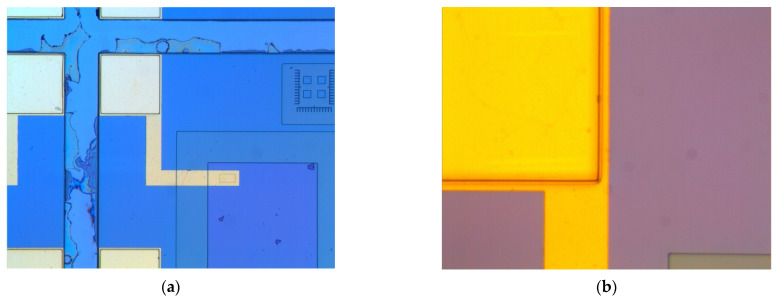
(**a**) Optical view of one external pad plus metallic trace to the source contact plus alignment sign and (**b**) details of the gold pad and gold trace to the source.

**Figure 6 biosensors-11-00497-f006:**
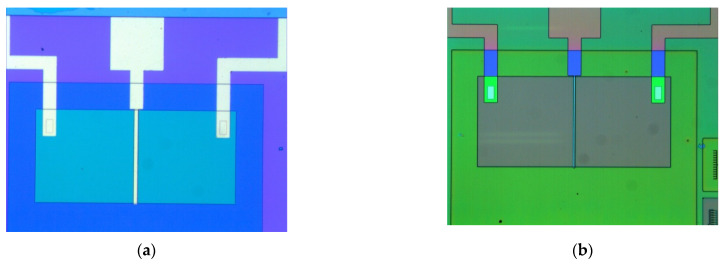
Image of the metallic contacts for the (**a**) reference MOS-FET (**b**) color-coded.

**Figure 7 biosensors-11-00497-f007:**
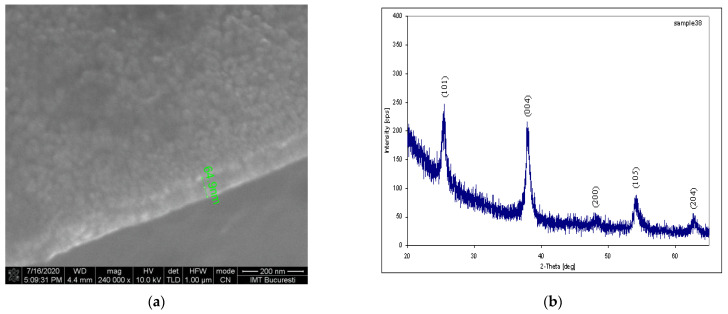
(**a**) SEM image for a TiO_2_ film converted to a nanostructured film at 450 °C in an N_2_ environment located over the FET gate. (**b**) XR spectrum for the same TiO_2_ film.

**Figure 8 biosensors-11-00497-f008:**
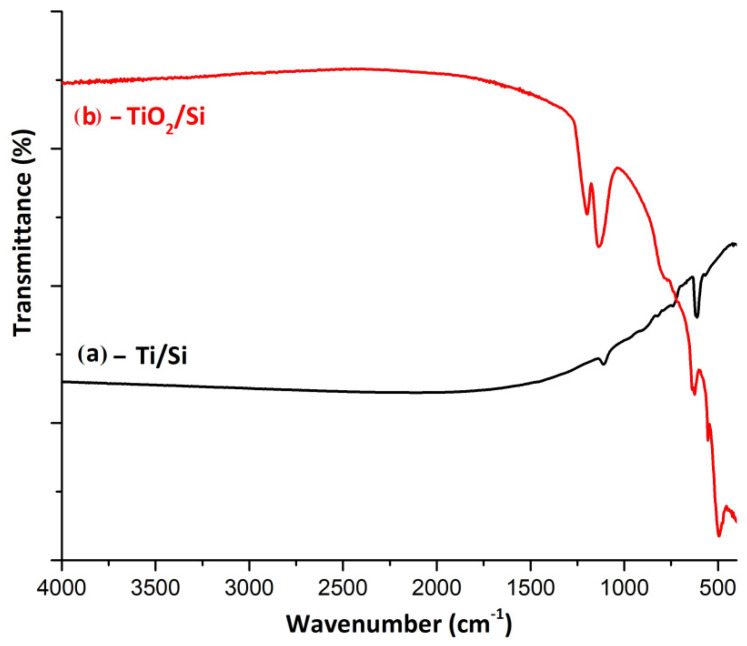
FTIR spectra for (**a**) a representative sample of titanium deposited on a silicon wafer and (**b**) for oxidized titanium film and heat treatment at 450 °C.

**Figure 9 biosensors-11-00497-f009:**
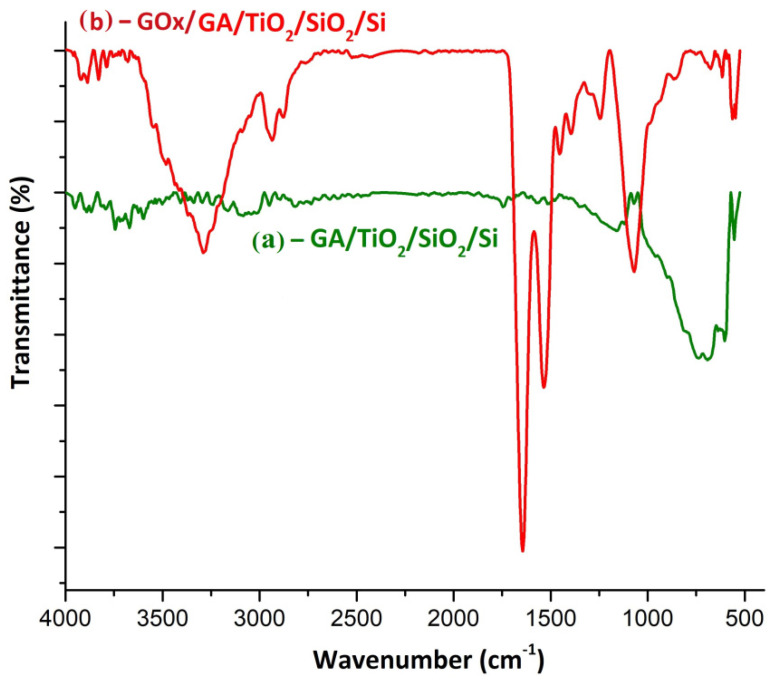
FTIR spectra for (**a**) a representative sample of GA deposited on nanostructured TiO_2_ film and (**b**) for GOx anchored to the GA–TiO_2_ probe.

**Figure 10 biosensors-11-00497-f010:**
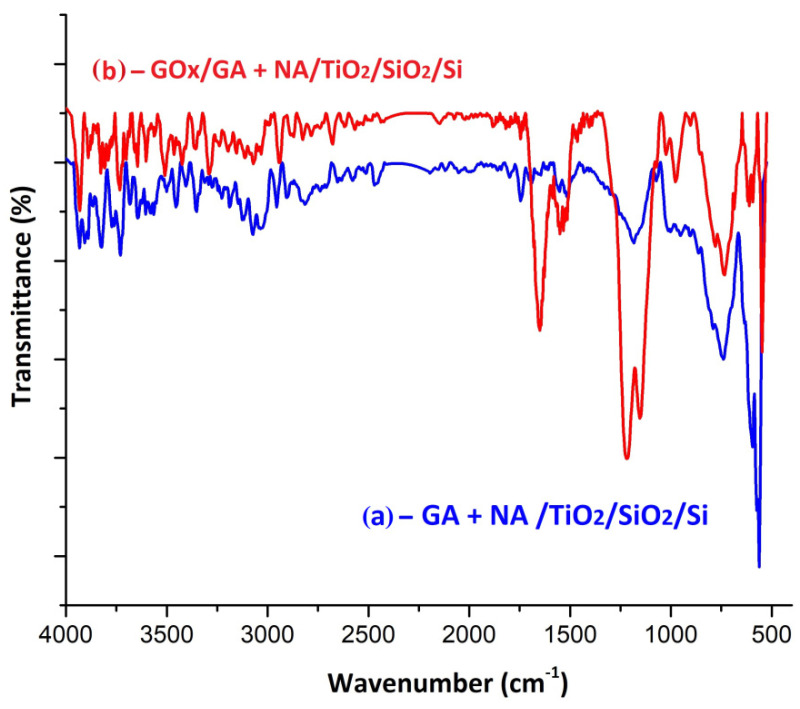
FTIR spectra for (**a**) a representative sample of GA deposited on a nanostructured TiO_2_ film and (**b**) for GOx anchored to a GA–TiO_2_ probe.

**Figure 11 biosensors-11-00497-f011:**
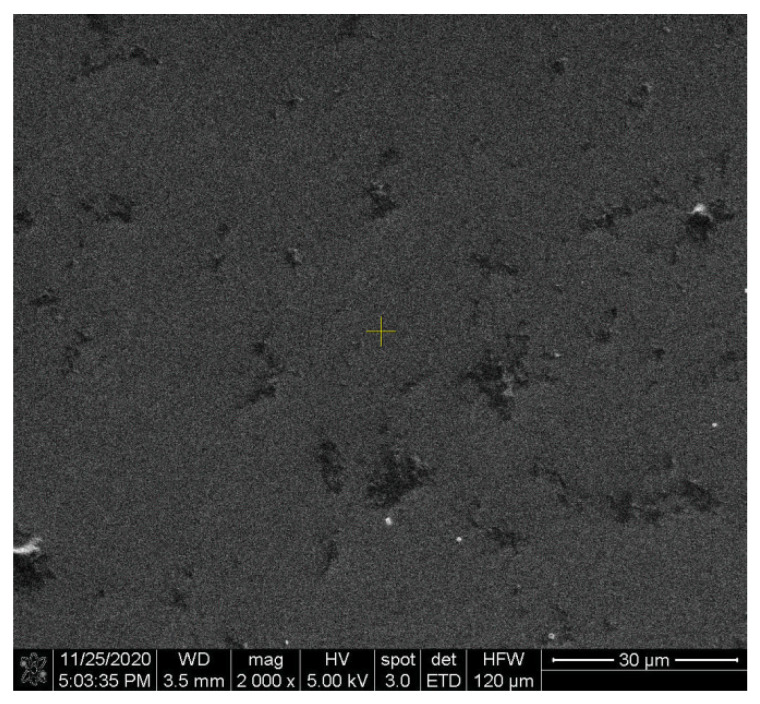
SEM image of the enzymatic membrane with GOx/(GA + NA)/TiO_2_/SiO_2_/Si from case (ii).

**Figure 12 biosensors-11-00497-f012:**
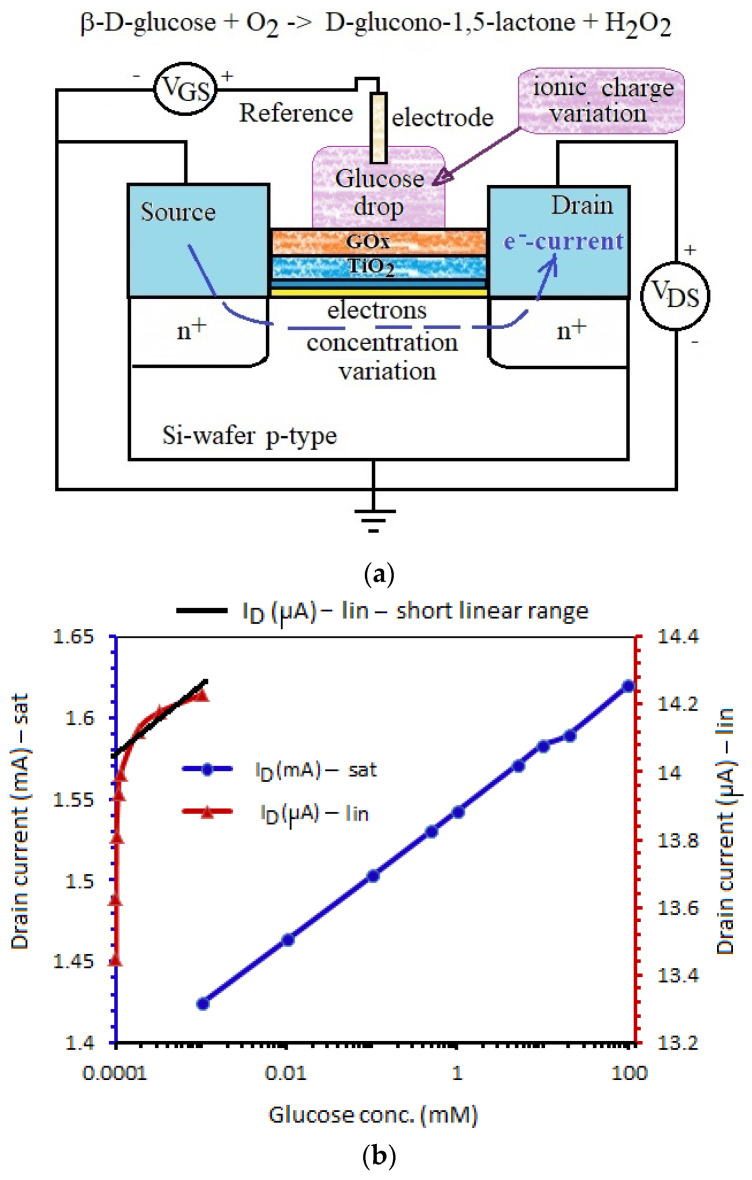
(**a**) The ENFET device functioning with a GOx enzyme. (**b**) The calibration curve of the glucose biosensor for saturation (sat) and linear (lin) work regimes of the ENFET transistor.

## Data Availability

PN2 National Projects 12095 internal database.

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
