# Peer review of "Optimized Technologies for Cointegration of MOS Transistor and Glucose Oxidase Enzyme on a Si-Wafer"

_biosensors, 2021, doi:10.3390/bios11120497_

Round 1

Reviewer 1 Report

Authors have improved a lot this manuscript. However, a little more details are required in the experimental section. Another question arises if the sample can be detected in different solvents, besides water, and its pH sensitivity.

Author Response

Authors reply to R1:

- English language suffered minor revisions and improvements in all sections. 

- More details are added in the experimental sections or little corrections:

- Old Fig. 3b didn't contain inside a visible explanation for the red curve, due to the Figure stretching. Now, we arranged area of Fig. 3b, to be visible both explanations inside picture - for red and for blue curves.

- In Section 2.4 we add more information about the reference electrode fabrication (see green text emphasized in yellow).

- Our co-author, Elena Manea gave us very precise corrections like:

Instead of: " In a preliminary stage, the wafers are cleaned in H2SO4 (90%) / H2O2 (30%) solution, .."

We correct to: "In a preliminary stage, the wafers are cleaned in H2SO4 (96%) : H2O2 (30%) (3:1) solution,"

Instead of: " After the implantation, subsequent annealing stages at maximum 1110 °C temperature, .."

We correct to: " After the implantation, subsequent annealing stages at maximum 1100 °C temperature, "

Instead of: " For metallic contacts, as Ti/Au (200nm/800nm) film is deposited onto the entire surface "

We corrected to: "For metallic contacts, Ti/Au (20nm/800nm) films are deposited onto the entire surface "

- to the end of Section 4. Calibration curve - we added new information, after Fig. 12 - if the sample can be detected in different solvents, besides water, and its pH sensitivity. (see emphasized text).

Some phrases were re-written to ensure a better clarity.

Authors thank again to Reviewers for their useful suggestions and pertinent observations that let us to improve our work.

Reviewer 2 Report

A significantly improved manuscript. Clear and concise. Dense of interesting observations well supported by the data.

Author Response

Authors reply to R2: English language suffered minor revisions. Some phrases were re-written to ensure a better clarity and minor revisions for Reviewer R1. 

Authors thank again to Reviewers for their useful suggestions and pertinent observations that let us to improve our work.

This manuscript is a resubmission of an earlier submission. The following is a list of the peer review reports and author responses from that submission.

Round 1

Reviewer 1 Report

The manuscript entitled “Optimized technologies for co-integration of MOS transistor 2 and glucose-oxidase enzyme on a Si-wafer” would like to describe a field-effect transistor wherein the gate electrode is modified with glucose oxidase. Glucose sensors attract attention due to their role in monitoring the physiological state of diabetics. Nevertheless, the manuscript does not reach the level for publication.

Comments:

  • The introduction does not provide the background that is necessary to understand the topic. The state of the art of glucose biosensors should be described and the authors should stress the strong point of their approach, by also describing the literature concerning transistors for the detection of glucose.
  • Methods are not adequately described. The name of all instruments should be reported in the manuscript. All the chemical reagents should be reported with the producer and the purity grade. How are the sensing measurements carried out? Different setup can be exploited for transistor measurements. For example, ISFETs use a reference electrode. No detail is reported in the manuscript. How is the electrochemical anodization performed? The setup of the measure should be described, with the electrical connection with the transistor. How is figure 13 obtained? What are the used equipment and the methodology?
  • The results do not show scientific soundness. They are focused on device fabrication by showing a lot of photo and IR spectra. All the images should be discussed in the text. Are figures 7 and 8 discussed in the text? No demonstration of transistor working is shown in the text. Output and transfer curves should be added to the work. The paragraph concerning sensing experiments is unclear. There are some data and no explanation about the experimental conditions and methodology for obtaining them. The only glucose calibration curve shows very low performance. For example, the signal recorded at a concentration of about 0.01 mM is very close to the signal recorded at a concentration value of about 3 mM. How can the sensor discriminate glucose concentration in this range? The authors do not report the errors associated with the calibration plot but they appear too big, also for a proof of concept.
  • The quality of the images is very low. It seems that the images are the simple instrument report without any stylistic elaboration. The IR spectra should be shown in only one image, to highlight the difference between the different stages of chemical modification. The quality of SEM images is very low. The pictures of the sensor do not help the readers to understand its structure and use.
  • Acronyms should be explained in the text.
  • Some details reported in the result and introduction sections (For example, the name of the instrument and the number of scans) should be moved in the methods section.

Reviewer 2 Report

Hi have to admit that I found this paper extremely hard to read.

The images are of poor quality and the the referee is really having some difficulties following the layout of the article.

Maybe the authors could start off with a general figure detailing what they are trying to achieve. The paper is full of messages and data but it is not clear which are the important ones.

For example, what is the point of the lineweaver-burk plots? Why the need to obtain Vmax and Km for GOX? Not perflectly known already?

And why the need of silica? Could not GOX be immobilised directly on the electrode using covalent Si-S or Si-O covalent chemistries? If the authors are not well equipped to perform reactions on Si-H surfaces, they could at least acknowledge this possibility, and referring for example to available reviews, Chem. Soc. Rev., 2011,40, 2704-2718

Reviewer 3 Report

Authors should consider rewrite all the manuscript and to follow the journal guidelines (i.e., formatting on images or tables, when is needed).  The manuscript looks like somewhat disorganized; the obtained results are randomly presented.  No comparison is observed with other basic biosensors.